



# Learning from a large-scale calibration effort of multiple lake models

Johannes Feldbauer[1], 🕊, Jorrit P. Mesman[2], 🕊, Tobias K. Andersen[3], and Robert Ladwig[4]

[1]TU Dresden, Dresden, Germany
[2]Uppsala University, Uppsala, Sweden
[3]National Institute of Aquatic Resources (DTU Aqua), Technical University of Denmark, Kgs. Lyngby, Denmark
[4]Department of Ecoscience, Aarhus University, Aarhus, Denmark
🕊These authors contributed equally to this work.

**Correspondence:** Johannes Feldbauer (johannes.feldbauer@tu-dresden.de) and Jorrit P. Mesman (jorrit.mesman@ebc.uu.se)

**Abstract.** Process-based hydrodynamic lake models are commonly used for simulating water temperature, enabling testing of different scenarios and drawing conclusions about possible water quality developments or changes in important ecological processes such as methane gas emissions. Even though there are several models available, a systematic comparison regarding their performance is missing so far. In this study, we calibrated four different one-dimensional hydrodynamic lake models for a

global dataset of 73 lakes to compare their performance in reproducing water temperature and estimated parameter sensitivity for the calibrated parameters. The models performance and parameter sensitivity showed a relation to the lake characteristics and model structure. No single model was the best, with each model performing better than the rest in at least some of the lakes. From the findings, we advocate the application of model ensembles. Nonetheless, we also highlight the need to further improve both weather forcing data, individual models, and multi-model ensemble techniques.

## 1 Introduction

The global rise in water temperatures in lakes and reservoirs (O'Reilly et al., 2015; Pilla et al., 2020) is affecting water quality and ecosystem services worldwide in multiple ways; e.g., promoting the formation of harmful cyanobacteria blooms (Huisman et al., 2018), modifying lake ice phenology (Knoll et al., 2019), affecting ecosystem functioning (Kattel, 2022), or increasing deep-water oxygen depletion (Jane et al., 2023). Water temperature is a "master variable" in aquatic biogeochemical cycling, involved in processes including kinetics of metabolism (Staehr et al., 2010) and greenhouse gas emissions (Audet et al.,

2017). Moreover, the vertical temperature structure controls mixing rates between water layers and modifies the position of organisms in the water column as well as the light and nutrient conditions they experience. As such, global future estimates of various water quality and ecological processes in inland waters should be based on an accurate model representation of present and future conditions of lakes temperatures and thermal structures that addresses the variability of lake characteristics worldwide. Recent continental and global-scale modeling efforts have presented convincing evidence of the large impact of

climate warming on lake temperatures (e.g. Woolway et al., 2021b; Golub et al., 2022). However, lake models can only be calibrated for comparatively few lakes for which in-situ, depth-resolved observations exist. Furthermore, there is a knowledge



gap on how model performance is affected by different lake-specific characteristics and how models could be parametrized based on the lake characteristics when applied on a global scale. At the moment it is common in global lake modeling studies

to apply models without lake- or region-specific calibration (e.g. Woolway et al., 2021a; Vanderkelen et al., 2020) and this adds considerable uncertainty to projections of climate change impacts on lake water temperatures.

Vertical one-dimensional (1D) hydrodynamic models are efficient tools to simulate water temperature dynamics for lakes in which the vertical density gradient is more pronounced than the horizontal one. Piccolroaz et al. (2024) gave an extensive review of the theoretical considerations for water temperature modeling across different spatial dimensions and noted the

frequent use of 1D models in climate simulations due to their low computational costs and adequate performance. Previous studies have indicated that optimal model parameter values may depend on certain lake characteristics, which could help to obtain more accurate fits in global applications. For instance, an application of the 1D physical lake model GLM (General Lake Model, see Hipsey et al., 2019) with a sensitivity analysis of nine model parameters across multiple lakes suggested that the sensitivity of a subset of parameters depended on characteristics such as lake depth, water transparency, and residence time

(Bruce et al., 2018). In a multi-lake application of the 1D physical model ALBM, Guo et al. (2021) highlighted the relationships between the relative influence of model parameters and lake characteristics such as latitude and lake depth. Extending beyond physical variables, Andersen et al. (2021) performed an extensive, global sensitivity analysis on the 1D coupled physical-biogeochemical model GOTM-WET in three Danish lakes and found that parameter sensitivity may be strongly linked to lake morphology. Regarding temperature simulations, biogeochemical components (such as light absorption by organic matter) may

be especially important in shallower lakes.

In this study, we applied four 1D physical lake models to a set of 73 lakes for which in-situ water temperature observations were available, as part of the Inter-Sectoral Impact Model Intercomparison Project (ISIMIP, Golub et al., 2022) using meteorological forcing from bias-corrected reanalysis data. The models were calibrated in a consistent manner, and we report on the overall model performance, highlight consistent patterns in model performance and parameter values, and assessed parameter

sensitivity. This study approach expands on previous studies through testing the sensitivity of multiple models simultaneously by applying an identical methodology for calibration and sensitivity analysis implemented over a larger number of lakes. Such an in-depth model evaluation on a global scale can:

1. Point towards systematic issues and biases of 1D physical lake models when forced by meteorological reanalysis data.

2. Reveal patterns in model performance driven by geographic location and/or lake characteristics.

3. Be used to test if an optimal model for specific lake types exists or, alternatively, advocate for an ensemble approach.

4. Identify a set of highly sensitive parameters for calibration.

This will expand our knowledge of the accuracy of water temperature modeling on a global scale, improve understanding of the relationship between lake characteristics and model parametrization thus giving advice for practitioners on how to best calibrate certain lake types, and potentially lead to a more accurate model application. As there is a growing interest in global

estimates of water quality and greenhouse gas emissions (e.g., Kakouei et al., 2021; Jansen et al., 2022; Jane et al., 2023),





which often rely partially on simulated water temperature and thermal structure, we need to ensure that the underlying global thermal information is as accurate as possible and that the level of uncertainty is known.

## 2 Methods

### 2.1 ISIMIP local lakes sector

ISIMIP — the Inter-Sectoral Impact Model Intercomparison Project — is a framework for consistently projecting the impacts of climate change across affected sectors and spatial scales (https://www.isimip.org; Frieler et al. (2024)). The Lake Sector is considering the impact of global warming on two categories of lakes: "local lakes" and "global lakes". The "local lakes" provides data for a set of 73 lakes where observed in-situ water temperature data and hypsographic information are available. This data was used for the present study (Table S1 in the Supplementary material, Golub et al. (2022)). The resolution (vertical

and temporal) of the observed data and the detail of the hypsograph varies per lake. No inflow and outflow data are available, so we assumed a constant water level throughout the simulation.

For the forcing of the models we used the GSWP3-W5E5 reanalysis dataset combining the GSWP (Kim, 2017; Dirmeyer et al., 2006) and the W5E5 datasets (Lange et al., 2021; Cucchi et al., 2020). The meteorological forcing, available at daily resolution, for each lake was extracted by the ISIMIP organizational team for the grid cells (spatial resolution: 0.5° by 0.5°)

in which each lake was located (Golub et al., 2022). The following meteorological variables were used to drive the lake simulations: air temperature, relative humidity, precipitation, shortwave radiation, longwave radiation, surface air pressure, and wind speed.

### 2.2 Lake clustering

In order to analyze the impact of lake characteristics (Table S2 in the Supplementary material) on model performance and

parameter sensitivity, we used K-means clustering to group the 73 lakes. Previous to clustering, we log-transformed elevation, mean depth, maximum depth and lake area and applied a z-score transformation. We created a silhouette plot to determine the optimal number of clusters, which was two. However, we decided to use five clusters instead as this gave a more meaningful representation of different lake types (Figure S1 in the Supplementary material).

### 2.3 1D physical lake models

Four vertical hydrodynamic models were used in this study to explore model sensitivity around climate change projections in lakes with varying algorithms and calculations regarding vertical temperature and heat transport: the two-layer (0.5D) model FLake (Mironov, 2008, 2005), the 1D integral energy model GLM version 3.1.0 (Hipsey et al., 2019), and the 1D turbulence-based models GOTM lake-branch version 5.4.0 (Burchard et al., 1999; Umlauf et al., 2005) and Simstrat version 2.4.1 (Goudsmit et al., 2002; Gaudard et al., 2019). The models were set up and run using the LakeEnsemblR R-package

(Moore et al., 2021) to standardize the approach. We refer to Piccolroaz et al. (2024) for detailed information regarding general




concepts in water temperature modeling. In this section we provide a summarized overview of the main differences in process description between these four models. The models were applied in an identical way, with the exceptions that FLake was used to simulate up to the mean depth instead of the maximum depth (in line with assumptions in the model), and that cloud cover was calculated from the meteorological variables using LakeEnsemblR functions for the GOTM model.

The vertical 0.5D (i.e. a box model, but with two separate boxes for upper and lower water layers) model FLake was originally designed for weather prediction studies, in which a large-scale climate model is coupled to multiple small-scale lake models. To achieve computational efficiency, FLake simulates the temperature dynamics of an upper completely-mixed layer and a thermocline layer (commonly also known as metalimnion), while neglecting temperature dynamics below the latter one (Mironov, 2005). Vertical temperature evolution itself is parametrized based on the self-similarity concept of the vertical

temperature profile (Kitaigorodskii and Miropolsky, 1970). This observed and theoretically-explained concept states that a dimensionless temperature profile in the thermocline can be replicated using a "universal" function of the dimensionless depth $\zeta$:

$$\frac{\theta_s(t) - \theta(z,t)}{\Delta\theta(t)} = \Phi_{\theta(\zeta)} \tag{1}$$

where $t$ and $z$ are the dimensions over time and depth, respectively. In equation 1 $\theta_s$ is the absolute temperature of the upper

completely-mixed layer, $\Delta\theta(t)$ is the absolute temperature gradient across the thermocline layer, and $\Phi_\theta$ is the "universal" function of the dimensionless depth. The dimensionless depth can be parametrized as:

$$\zeta = \frac{z - h(t)}{\Delta h(t)} \tag{2}$$

where $h(t)$ is the depth of the upper, completely-mixed layer, and $\Delta h(t)$ is the depth difference between the mixed layer depth and the bottom of the metalimnion. Note that in this study, we set the bottom metalimnion depth to each lake's mean

depth. Applying this concept to temperature evolution, FLake parametrizes both layers (upper completely mixed and thermocline layer) as:

$$\Theta = \begin{cases} \theta_s, & if\, 0 < z < h \\ \theta_s - (\theta_s - \theta_b)\,\Phi(\zeta), & if\, D_{lake} \le z \le h \end{cases} \tag{3}$$

where $D_{lake}$ is the maximum depth (Mironov, 2005). Similar to the other models, the upper completely-mixed layer receives the energy fluxes from the atmosphere:

$$h\frac{d\theta_s}{dt} = \frac{1}{\rho_w c_w}(Q_s + I_s - Q_h - I(h)) \tag{4}$$

where $\rho_w$ is water density, $c_w$ is heat capacity, $Q_s$ is the turbulent heat flux at the surface, $I_s$ is the surface short-wave radiative flux, $Q_h$ is the heat flux from the bottom to the upper layer, and $I(h)$ is the radiative short-wave flux through the



water column (Mironov, 2005). We can state the sum of these individual heat fluxes as the net heat flux exchange $H_{net}$. Although FLake's numerical implementation combines empirical formulations with physical processes, it has demonstrated a
good performance for surface water temperature modeling as well as ice phenology investigations (e.g. Mallard et al., 2014) and is commonly applied to global studies (Woolway and Merchant, 2019).

GLM, GOTM and Simstrat are vertical 1D hydrodynamic models in which temperature evolution is quantified at every time step over a vertical grid. Conceptually, the models differ regarding how the vertical grid is configured as GLM applies a flexible structure, whereas the others use a fixed grid with the possibility of refinements. Nonetheless, all three models are based on the
vertical water temperature equation, which — in its general form — can be stated as:

$$\frac{\partial T}{\partial t} = \underbrace{-\frac{1}{\rho c_p}\frac{\partial I}{\partial z}}_{(1)} + \underbrace{\frac{H_{sed}}{A\rho c_p}\frac{\partial A}{\partial z}}_{(2)} + \underbrace{\frac{S_T}{\rho c_p}}_{(3)} + \underbrace{\frac{1}{A}\frac{\partial}{\partial z}\left(AD_z^T\frac{\partial T}{\partial z}\right)}_{(4)} \tag{5}$$

in which the change of temperature $T$ over time depends on four terms on the right hand-side: (1) the internal heat generation due to short-wave solar radiation $I$, (2) a geothermal heat flux $H_{sed}$ that acts over an area $A$, (3) an internal heat source term $S_T$, (4) and a turbulent diffusive term that includes the eddy-diffusivity coefficient $D_z^T$ (Piccolroaz et al., 2024). The layer next
to the atmosphere-water interface receives a net heat flux exchange similar to the one described in equation 4, where $H_{net}$ is the sum of radiative and turbulent heat fluxes:

$$\rho_w c_p (D_z^T \frac{\partial T_z}{\partial z})\bigg|_{z=s} = H_{net} \tag{6}$$

where $T_z$ is the water temperature of the layer adjacent to the atmosphere-water interface at the surface depth $s$.

The main difference between GLM and both GOTM and Simstrat, is how they simulate the turbulent diffusive transport.
GLM applies a combination of empirical and physical relationships that use the available external turbulent kinetic energy (TKE) to calculate the thickness of a completely-mixed surface layer (for more information about integral energy models see Ford and Stefan, 1980). For this, mixing in a surface mixed layer is calculated by comparing the available external energy to the potential energy of the water column that is needed to lift up denser water from below a completely-mixed layer into a newly formed mixed layer until the TKE is no longer sufficient for further mixing (Hipsey et al., 2019). Below the depth of
this surface mixed layer, a parametrization for the eddy diffusivity coefficient in relation to water column stability is used to calculate diffusive transport:

$$D_z^T = \frac{C_{HYP}\varepsilon_{TKE}}{N^2 + 0.6\,k_{TKE}^2\,u_*^2} \tag{7}$$

where $C_{HYP}$ is a constant coefficient for the mixing efficiency, $\varepsilon_{TKE}$ is a simplified approximation of turbulent dissipation rate based on the dissipation by inflows and wind, $N^2$ is the squared buoyancy frequency, $k_{TKE}$ is the turbulence energy con-
taining wavenumber, and $u_*$ is the wind shear velocity (Weinstock, 1981). The buoyancy frequency (Brunt-Väisälä frequency) quantifies local stability to vertical displacements as:



$$N = \sqrt{\frac{g}{\rho} \frac{\partial \rho}{\partial z}} \tag{8}$$

where $g$ is gravitational acceleration.

Simstrat and GOTM are turbulence-based models that apply a two-equation turbulence model to compute the quantities of
the production, transport and dissipation rates of TKE. Here, we highlight the $k - \varepsilon$ two-equation turbulence model which is
implemented in both models (Burchard et al., 1999; Goudsmit et al., 2002):

$$\frac{\partial k}{\partial t} = \frac{1}{A} \frac{\partial}{\partial z} \left( A D_z^k \frac{\partial k}{\partial z} \right) + P + B - \varepsilon \tag{9}$$

$$\frac{\partial \varepsilon}{\partial t} = \frac{1}{A} \frac{\partial}{\partial z} \left( A D_z^\varepsilon \frac{\partial \varepsilon}{\partial z} \right) + \frac{\varepsilon}{k} \left( c_{\varepsilon,1} P + c_{\varepsilon,3} B - c_{\varepsilon,2} \varepsilon \right) \tag{10}$$

where $D_z^k$ and $D_z^\varepsilon$ are the turbulent diffusivities of TKE and TKE dissipation, respectively, $P$ is the TKE production due
to shear, and $B$ is the production and dissipation of TKE related to buoyancy (Rodi, 1984). $c_{\varepsilon,1}$, $c_{\varepsilon,2}$ and $c_{\varepsilon,3}$ are empirical
constants. We can compute the eddy diffusivity coefficient $D_z^T$ as a function of the turbulence kinetic energy $k$ and dissipation
rate $\varepsilon$:

$$D_z^T = \frac{c_\mu}{\sigma_t} \frac{k^2}{\varepsilon} \tag{11}$$

where $c_\mu$ is an empirical coefficient, and $\sigma_t$ is the turbulent Prandtl number.

Simstrat further employs an empirical seiche excitation and damping model to improve the representation of internal seiches
in transport processes (Goudsmit et al., 2002). Here, seiche movement can produce additional TKE, $E_{seiche}$, inside the water
column with the intention to provide a more realistic simulation of vertical transport due to bottom boundary mixing as seiche
motion damping is an energy source below the mixed layer:

$$\frac{dE_{seiche}}{dt} = \underbrace{\alpha A_0 \rho_{air} c_{10} (u_{10}^2 + v_{10}^2)^{3/2}}_{PW} - \underbrace{C_{Deff} A_0 V^{-3/2} \rho_0^{-1/2} E_{seiche}^{3/2}}_{LS} \tag{12}$$

where $PW$ is energy production, $LS$ is energy loss, $\alpha$ is a model parameter to describe the wind energy fraction that is
transferred to the seiche motion, $c_{10}$ is the drag coefficient, $u_{10}$ and $v_{10}$ are velocity components of wind speed measured at
10 m above water surface, and $C_{Deff}$ is the effective bottom friction coefficient (Goudsmit et al., 2002). We note that similar
algorithms, designed to improve vertical mixing dynamics below the epilimnion, exist also in other models, including integral
energy models, i.e., the turbulent benthic boundary layer mixing algorithm by Yeates and Imberger (2003), but are not — to
the best of our knowledge — implemented in GLM and GOTM.

An additional structural difference between the models is their process description of the treatment of attenuation of short-
wave radiation, especially the non-visible near-infrared light (NIR) and the visible parts of short-wave radiation. FLake does





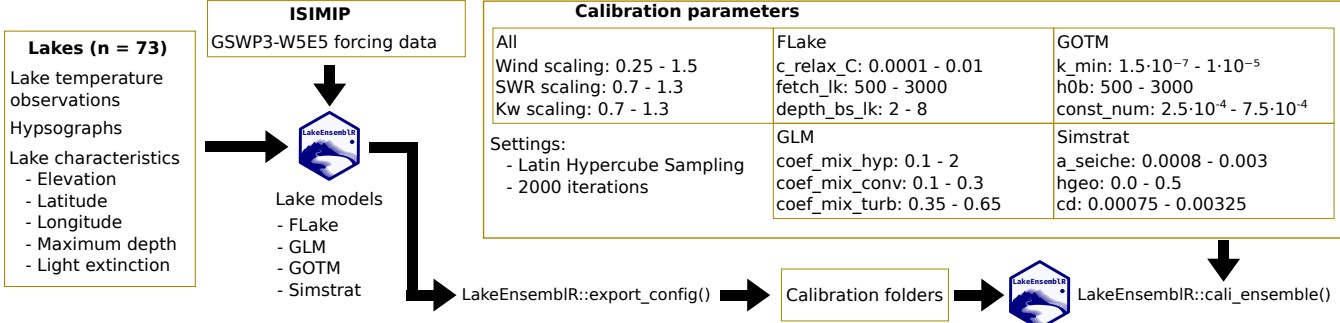

**Figure 1.** Workflow of the calibration, for a description and units of the calibrated parameters see Table 1. The light extinction coefficient (Kw) was calibrated 30% around the default value for each specific lake.

not distinguish between these parts of the light spectrum, and applies the Beer-Lambert law for light attenuation with depth (see also Stepanenko et al., 2014, for a more detailed analysis), although the model can be parametrized to consider a set

of different wavelength bands with variable attenuation coefficients (Mironov, 2005). GLM has the option to apply the Beer-Lambert law for only the photosynthetically active fraction, while the NIR and ultraviolet bandwidths are attenuated directly in the layer adjacent to the atmosphere-water interface (Hipsey et al., 2019). A second option in GLM uses the algorithm by Cengel and Ozisk (1984) to simulate light penetration of individual bandwidth fractions. However, in this study the first option was applied which treats 45% of the incoming short-wave radiation as PAR which subsequently is attenuated in the layers

below the atmosphere-water interface. Similarly, GOTM was configured to have a separate depth-specific attenuation for the visible and non-visible light fractions. In this study, the incoming short-wave radiation was split into non-visible and visible fraction with 55% and 45%, respectively. The light extinction coefficient for non-visible light was set to $2 \text{ m}^{-1}$. Although Simstrat does not split light into separate fractions, it uses a parameter to absorb a fixed fraction of short-wave radiation (in this study set to 30%) in the uppermost water layer, in effect realizing a similar impact of fast absorption of a part of the

solar energy near the atmosphere-water interface (see also Gaudard et al., 2019). This highlights that potentially more heat got absorbed in the layer adjacent to the atmosphere-water interface in the GLM, GOTM, and Simstrat simulations than in the FLake simulations.

## 2.4 Calibration workflow

The workflow (Figure 1) to calibrate the models for the 73 lakes is described in the following section. For each lake we gathered

the available data from ISIMIP: observed water temperatures, lake hypsography, lake location (elevation, coordinates), and light extinction (or Secchi disk depth data to derive light extinction). Observed water temperature data with subdaily resolution were averaged to daily mean values. If no data on the light extinction was available, we estimated it from Secchi disk depth (Koenings and Edmundson, 1991). If no Secchi disk depth was available as well, we estimated it from the maximum lake depth (Håkanson, 1995). We then formatted the ISIMIP data to a pre-defined standard format, from which the LakeEnsemblR



**Table 1.** Description of the calibrated parameters. For the range of the parameters see Figure 1

| Parameter | Unit | Description | Model |
|---|---|---|---|
| wind_speed | - | Scaling factor for wind speed | All models |
| swr | - | Scaling factor for incoming shortwave radiation | All models |
| Kw | 1/m | Scaling factor for estimated light extinction | All models |
| c_relax_c | - | Constant in relaxation equation of shape factor | FLake |
| fetch_lk | m | Typical wind fetch | FLake |
| depth_bs_lk | m | Depth of thermally active layer in bottom sediments | FLake |
| k_min | $m^2/s^2$ | Minimum turbulent kinetic energy | GOTM |
| h0b | m | Physical bottom roughness length | GOTM |
| const_num | $m^2/s$ | Constant eddy diffusivity | GOTM |
| coef_mix_hyp | - | Mixing efficiency of hypolimnetic turbulence | GLM |
| coef_mix_conv | - | Mixing efficiency of convective overturn | GLM |
| coef_mix_turb | - | Mixing efficiency of unsteady turbulence effects | GLM |
| a_seiche | - | Fraction of wind energy that goes to seiche energy | Simstrat |
| hgeo | $W/m^2$ | Geothermal heat flux | Simstrat |
| cd | - | Bottom drag coefficient | Simstrat |

package (Moore et al., 2021) generated model-specific forcing and configuration files. We used four hydrodynamic lake models included in LakeEnsemblR (GLM, GOTM, Simstrat, and FLake) which are described in Section 2.3.

Finally we ran the calibration using a latin hypercube approach (see e.g. Mckay et al., 2000). Here, we chose 6 parameters for each model: three model-specific parameters, and three scaling factors (for wind speed, incoming shortwave radiation, and the estimated light extinction coefficient, respectively) (Table 1). For the model-specific parameters, we chose parameters that

are commonly used to calibrate these models, based on literature (see Moore et al., 2021) and personal communication with model users. We sampled and ran the four models for 2000 parameter sets, and for each of the parameter sets we calculated four performance metrics for water temperature profiles: Root mean squared error (RMSE), Nash-Sutcliffe model efficiency (NSE), Pearson correlation coefficient (R), and mean error (bias).

## 2.5   Global sensitivity analysis

Based on the sampled parameter sets and the calculated performance metrics, we performed a Delta Moment-Independent sensitivity analysis (Plischke et al., 2013; Borgonovo, 2007) for each performance metric per lake per model, using the Python library SALib (Iwanaga et al., 2022; Herman and Usher, 2017). The analysis calculates the two sensitivity measures $\delta$ and $S1$, where the total-order delta moment-independent measure $\delta$ considers all interactions between parameters, and variance-based first-order sobol index $S1$ only considers each parameter's specific influence (Plischke et al., 2013; Borgonovo, 2007). Applying

both a moment-independent method ($\delta$) and a variance-based first-order method ($S1$) increases the robustness of estimating





the inference of which parameters are influential (Borgonovo et al., 2017). In addition to the six calibrated parameters, we included a dummy parameter that had no influence on the model output in the sensitivity analysis, which we sampled from a uniform distribution ranging from 0 to 1. In theory, this dummy variable should have a sensitivity of zero, but due to the numerical approximation of the sensitivity measures it can have small non-zero values. This can be used to approximate the
error of estimating sensitivity indices and thereby avoid classifying non-influential parameters as influential. This approach has been used in previous studies (e.g. Andersen et al., 2021; Khorashadi Zadeh et al., 2017). A resample size of 100 was used to compute confidence intervals on both SA metrics. To provide an estimate of potential parameter interactions, we additionally calculated the interaction indicator $S_{interaction}$ (Borgonovo et al., 2017; Saltelli et al., 2000) that describe the fraction of model output variation apportioned by interactions:

$$S_{interaction} = 1 - \sum_{i=1}^{k} S_i \tag{13}$$

where ($S_i$) is the first-order variance-based sensitivity measure ($S1$) of parameter $i$ out of $k$ tested parameters.

## 3    Results

### 3.1    Model performance

The single best performing model (out of the four applied models) for each lake reproduced observed water temperatures well
for all 73 lakes, with a median RMSE of 1.2 °C and a median R of 0.98 (Figure 2). The variation in error metrics between best performing model for each lake was rather small, e.g. a standard deviation of 0.5 °C or less in RMSE (Figure S2 in the Supplementary material). Simstrat performed the best in most lakes in terms of RMSE, R, and NSE, while GLM performed best in most lakes for bias (Figure 2). However, all four models outperformed the others in at least some of the lakes. In over 90% of all lakes at least two different models performed best for different metrics.
Following the cluster analysis, we classified the lakes into five cluster. We visually compared the characteristics of the cluster (Figure S1 in the Supplementary material) and characterized them according to their most noticeable features: "deep" (n = 3), "medium temperate" (n = 25), "small temperate" (n = 32), "large shallow" (n = 4), and "warm" lakes (n = 9) (Figure 3). Model performance was comparable between the cluster, although the "deep" lakes had a lower RMSE while "medium", "small temperate", and "large shallow" lakes performed best in terms of NSE and R (Figure S3 in the Supplementary material). When
considering the four models separately, the overall better performance of Simstrat was mostly due to its better performance in the "deep" and "medium temperate" lakes, compared to the other models. In the other three cluster the four models performed similarly (Figure S5 in the Supplementary material).
We calculated the ensemble mean by taking the arithmetic mean of the four models for each time step and depth individually. We then tested this as an additional predictor for water temperature and calculated its performance in terms of RMSE. For the
majority of lakes, the ensemble mean performed better than any single model (Figure S4 in the Supplementary material).





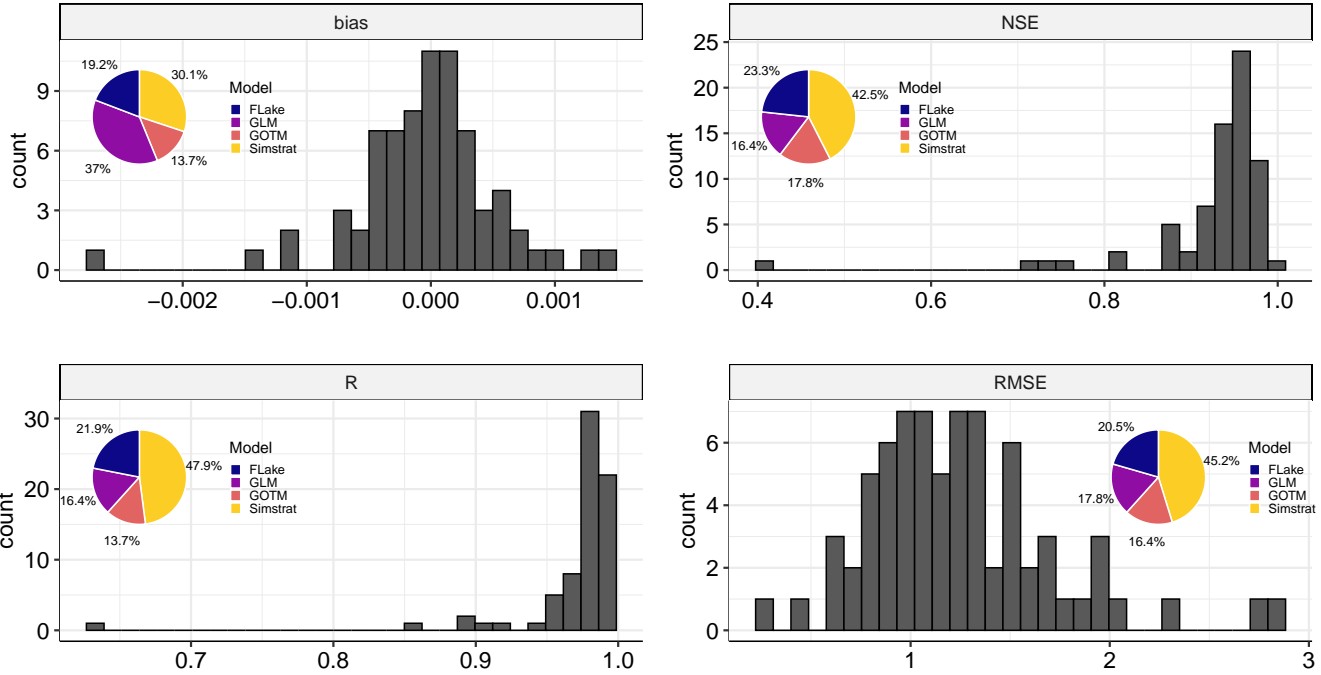

**Figure 2.** Distribution of the six evaluated performance metrics for the single best performing model over the 73 lakes. The pie charts show how often the different models performed best per lake and metric. The unit of RMSE and bias is K

This is especially visible in the "medium temperate", "small temperate", and "large shallow" lakes where the ensemble mean performed best for the majority of lakes. The cases where the ensemble mean did not perform better than each single model were often lakes in which a single model performed notably better or worse than the other three models.

Looking at the distribution of the model error in terms of RMSE over the water column depth (Figure 4 A), we can see that for the "medium temperate" lakes, Simstrat performed better over all depths. In the "deep" lakes, FLake performed considerable worse than the other three models, especially at intermediate depths. For the other three models, the error was increasing towards the surface. For all four models in the "large shallow" lakes the error was larger at the surface, while for the "warm" lakes the error was largest in the intermediate depths.

From the observed water temperatures, we calculated the thermocline depth and then chose the simulation–observation pairs closest to that depth to estimate the RMSE at the thermocline (Figure 4 B). For the "large shallow" lakes, no thermocline could be calculated. Simstrat performed best at the thermocline depth for "deep", "medium temperate", and "warm" lakes, whereat the performance in "deep" and "medium temperate" lakes was about 0.5 K better, while for "warm" lakes it was only about 0.1 K lower than the next best model. For "small temperate" lakes GLM performed better with an median RMSE about 0.3 K lower than the next best model. FLake performed poorest at the thermocline depth in all lake cluster.





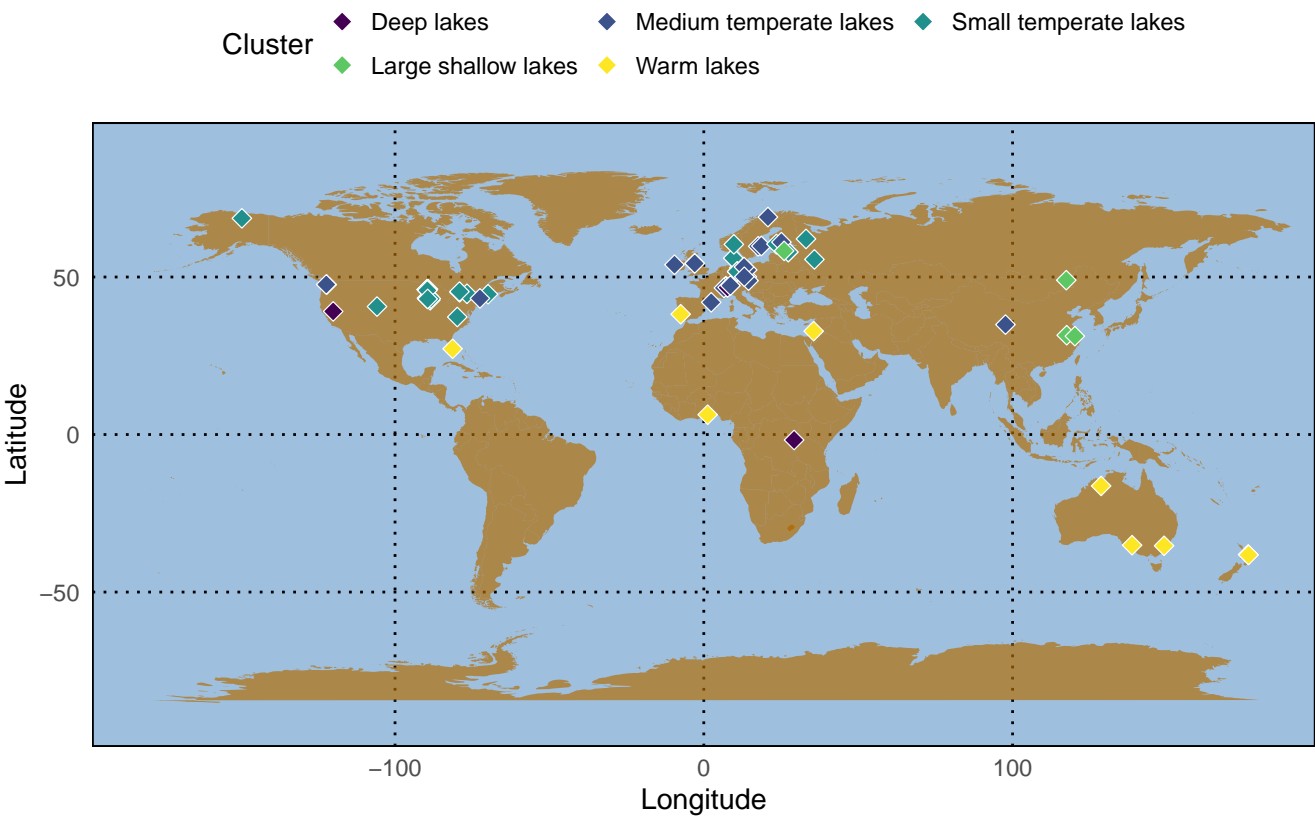

**Figure 3.** Map showing the locations and grouping derived by K-means clustering of the 73 lakes included in the study.

## 3.2 Parameter sensitivity

From the calibration runs using the latin hypercube approach, we calculated density-based total order measure $\delta$ and variance-based first-order measure $S1$ for each combination of models, performance metrics, and lakes (Figure 5). We saw similar ranking of most influential model parameters on most combinations of models, performance metrics, and lakes for $\delta$ and $S1$. For almost all lakes, the same $3-4$ parameters were classified as sensitive: the scaling factors for wind speed, short wave radiation, and light extinction as well as `k_min` for GOTM. Moreover, 1 or 2 of these parameters accounted for more than 75% of the sum of the sensitivity measures for most lakes (Figure S6 in the Supplementary material). Most often these were meteorological scaling factors, that are not model specific, with the exception of GOTM, in which `k_min` was most sensitive. Additionally, the light extinction coefficient and other model-specific parameters appeared to be sensitive in a couple of lakes (Figure 5), but to a lesser degree.

For most models and performance metrics, interaction effects accounted for less than 20% of variation in model performance, though interactions were relevant for specific models and lake groups (Figure 6). For instance, interactions were relevant for



**Figure 4.** Depth distribution of the root mean squared error (RMSE) for the models and lake cluster (A) and boxplots of RMSE at the thermocline depth (B). The depth was normalized to the depth of the deepest measurement (0 being surface and 1 being the deepest point) and then binned in steps of 0.1. The points represent the median RMSE over all profiles and the error bars the 25% to 75% quantiles. If water temperatures deeper than 3 meter were unavailable, no thermocline was calculated.





GLM modeling "deep" lakes and to a lesser degree GLM and Simstrat modeling "large shallow" lakes. In contrast, increased parameter interactions was observed for FLake, especially for NSE and RMSE, for all lake cluster except "deep" lakes. We highlight that especially in lakes with shorter time series of observed water temperature data, the interaction measure was larger
(Figure S7 in the Supplementary material). Interactions were low for bias for all models and lake cluster.

### 3.3 Distribution of best parameter values

Looking at the parameter values from the best performing parameter sets, the optimal meteorological scaling factors differed between models. Especially GOTM showed a different behavior from the other models, with lower wind speed scaling factors and a higher shortwave radiation scaling factor (Figure 7). The lake cluster also differed in the optimal scaling factors, though
their effects seemed model-specific. Differences in extinction factor scaling were less clear than the meteorological scaling factors, but GLM preferred a higher extinction factor in "large shallow" lakes, and FLake a higher extinction factor in "deep" and "medium temperate" lakes. Most model-specific parameters had a low sensitivity, but some still showed markedly different behavior between cluster (Figure S8 in the Supplementary material). The single model-specific parameter with high sensitivity, GOTM's k_min, had distinctly lower values in "small temperate" lakes. Both for the scaling factors and the model specific
parameters, we saw that depending on which performance metric was used to select the best parameter set, the outcome was different (see Figure S9 in the Supplementary material).

### 4 Discussion

Using a standardized and computationally efficient calibration approach (2000 model runs per model and lake) we were able to reproduce water temperature to a sufficient accuracy for 73 lakes across the globe. For 95% of the lakes, the single best
performing model had an RMSE below 2 °C with a median of all performing models at 1.2 °C. Model-specific performance (Table S3 in the Supplementary material) naturally showed higher error values but remained below 2 °C for most lakes and models. Compared to a previous ISIMIP simulation round, the performance in terms of median RMSE was similar (ISIMIP 2b, Golub et al., 2022), although in our simulation, GLM, GOTM, and Simstrat performed slightly worse and FLake slightly better. Possible reasons for this could be the different meteorological forcing, the composition of lakes, and a different calibration
approach. In comparison to two other multi-lake applications of gridded meteorological data, our calibration performed similar (ALBM, Guo et al., 2021) or better (GLM, Read et al., 2014) in terms of RMSE. In over 40% of all lakes, the ensemble mean performed better than any single model in terms of RMSE. Similar to previous studies using an ensemble framework (e.g. Feldbauer et al., 2022; Ladwig et al., 2023), it seems that the ensemble mean is a good predictor for water temperature dynamics. However, when using a larger global data set, we showed that for a subset of lakes using the simple arithmetic
average as an ensemble mean did not increase performance. As in many of these cases a single model performed notably better or worse than the other ensemble members, so a step forward could be to use other averaging techniques to make better use of the ensemble simulation. Such approaches already exist in other fields, like the reliability ensemble averaging method (REA) for climate simulations (Giorgi and Mearns, 2002).





**Figure 5.** Boxplot of the two calculated sensitivity measures for each parameter of the four models and for the four calculated performance metrics over all lakes.





**Figure 6.** Boxplot of the interaction measure of first order sensitivity metric for the four models and performance metrics over all lakes.

Model performance showed a distinct pattern over the five lake cluster: when looking at RMSE, general model performance
in "deep" lakes was better, while in "large shallow" lakes it was worse compared to the other cluster. But, both "deep" and "warm" lakes showed poorer model performance when considering NSE (Figure S3 in the Supplementary material). We attribute the model performance in "deep" lakes (n = 3) to the low variation in the deep water temperatures, which the models could approach closely (low RMSE), whereas the relatively small temporal variations were harder to simulate (i.e. poorer performance in terms of NSE). The reduced model performance in terms of RMSE for "large shallow" lakes (n = 4) was likely
due to the intense interaction with the atmosphere (worsened by the use of gridded instead of locally observed meteorological data), while the lower NSE in "warm" lakes (n = 9) can be explained by the reduced seasonality in weather forcing data, and thus a harder-to-achieve high performance in metrics relying on temporal trend. Other performance differences between lake cluster, such as in bias or any differences between the two largest cluster ("small temperate" and "medium temperate" lakes)





**Figure 7.** Distribution of the wind speed, shortwave radiation (swr), and light extinction (Kw) scaling factors, faceted by model and archetype. The light extinction scaling factors are normalised to each lake's default light extinction value. The optimal values are determined only based on RMSE.

were marginal. These two largest cluster covered 78% of the lakes in the dataset, which is in line with the higher presence of

temperate lakes in the ISIMIP dataset. However, the unequal division of lakes over the cluster does skew the comparison as drawing conclusions regarding differences in other cluster are based on a lower sample sizes.

To discuss how individual model performance is related to the underlying equations and design, we first need to acknowledge limitations of this analysis: (a) parameter selection was limited and had identical ranges across models, which could cause a bias for models that would need specific adjustments during calibration, and (b) we neglected any inflows and out-

flows. Observed water temperature fluctuations caused by entrainment or withdrawal could be apparent in the training data and models could replicate them by manipulating other processes (internal shear or mixing), thereby neglecting the actual hydrodynamic flow processes which caused the above-mentioned temperature fluctuations. We note that the level of complexity



in the process formulations for inflows and outflows varies across the models. Putting these caveats due to the standardized methodology aside, 1D hydrodynamic models have an improved performance compared to the single 0.5D model, like FLake,
for "deep" and "medium temperate" lakes. Here, all 1D lake models better replicated water temperatures in the surface layers (relative depths up to 0.75 and about 0.5 for "deep" and "medium temperate lakes", respectively, Figure 4 A) underscoring that their respective algorithms, wind-induced mixing in GLM and computation of TKE in GOTM and Simstrat, outperforms the shape assumptions that underlie FLake to replicate depth-specific near-surface water temperature dynamics. Additionally, their higher light extinction near the atmosphere-water surface due to attenuation of non-visible light could also be a factor
in their improved depth-specific simulation of water temperature in "medium temperate" and "deep" lakes. Below the epilimnion, at the thermocline, Simstrat outperforms the other models (Figure 4 B) in "deep" and "medium temperate" lakes. This underscores the importance of accounting of energy sources below the epilimnion. We assume that Simstrat's seiche excitation and damping parametrization has more accurately simulated the availability of TKE at these metalimnetic depths, which were not reached by wind shear stress originating from the atmosphere-water interface. This emphasizes the importance of
implementing deep-water mixing algorithms in 1D hydrodynamic lake models to account for mixing at intermediate depths, which are usually characterized as quiet regarding small-scale turbulent fluxes (Wüest and Lorke, 2003). In the hypolimnion, models performed similarly, with Simstrat only producing slightly better replications of deep-water temperature in "medium temperate" lakes.

For the calibration of the lake models we took an approach commonly used in applied studies where scaling factors for
wind speed and shortwave radiation, the extinction coefficient, and a few model-specific parameters are calibrated (e.g. Ayala et al., 2020; Weber et al., 2017). Additionally, we used the output of the calibration to conduct a global sensitivity analysis of the calibrated parameters. We selected the model-specific parameters and the ranges for all parameters based on previous studies and expert knowledge, but acknowledge that this approach is somewhat limited compared to an extensive sensitivity analysis including all model parameters. However, to our knowledge there are only a few studies looking at the sensitivity of
the parameters of the used models (e.g., GLM - Bruce et al., 2018; GOTM - Andersen et al., 2021) and even those did not include all model parameters. Moreover, the model performance of all four investigated performance metrics was comparable to similar studies (e.g. Golub et al., 2022), despite using only a selection of parameters for the calibration. The sensitivity analysis revealed that for most lakes and models, the most sensitive parameters were the scaling factors. So, it could be reasoned that for similar applications only calibrating the scaling factors could be sufficient. The clear exception here is GOTM, for which
minimum turbulent kinetic energy level (k_min) showed to be highly sensitive. In fact, k_min was so important that it could dominate the other scaling factors leading to different overall patterns in the calibrated parameters with lower values for the wind speed scaling across all lakes, compared to the other models (Figure 7). This warrants future caution when calibrating k_min as this parameter, which directly manipulates background turbulent kinetic energy and therefore turbulent transport, is highly sensitive. A way forward to address this could be to use local field measurements to restrict the range of estimates for
k_min.

Especially the range for the wind speed scaling which we used in the calibration was quite large (0.25 - 1.5) and even with this range for some of the lakes the best performing estimates are located close to the limits. An explanation for this large




range of scaling factors is that we used forcing from bias-corrected (to global data sources, not data measured above the lakes, Lange (2019)) reanalysis data with a grid size of 0.5°x0.5°. Local wind fields can have large variations and especially for lakes,

sheltering plays an important role, as lakes are by definition located in depressions in the landscape. Simultaneously, larger lakes can act as smoother surfaces with higher near-surface wind speeds compared to surrounding areas. Due to local variability and limitations in data products, there is still potential for enhancement in model quality of local wind speed (Tan et al., 2024). The use of daily-aggregated wind speeds also requires caution, as the mechanic energy transferred to the water is a cubic function of wind speed (Wüest et al., 2000), and therefore averaging of the measured wind speed can lead to underestimation

of mixing. The large range for the wind speed (and shortwave radiation) scaling factors were probably partly responsible for their high sensitivity. In a setting with locally observed meteorological forcing data, the model-specific parameters might become more influential, if meteorological forcing variables can be more constrained.

We highlight that both sensitivity metrics and calibrated parameter values were strongly influenced by the chosen performance metrics (see e.g. Figure 5 and Figure S9 in the Supplementary material). This means that depending on which perfor-

mance metric is chosen, the model configuration would be different (except for RMSE and NSE, which will lead to the same set of parameters). Therefore, it is important to choose the model performance metric with care as they capture different aspects of the performance (see e.g. Jachner et al., 2007). For a more thorough assessment of the choice of performance metrics, model validation at multiple levels of complexity could be performed (Hipsey et al., 2020).

Interaction between parameters was larger for FLake specifically, and for GLM in "deep" and for GLM and Simstrat in "large

shallow" lakes (Figure 6). For the lakes with high interaction measures, we found interdependence of two or more parameters, most notably wind speed scaling and shortwave radiation scaling, in some cases also the light extinction factor or model-specific parameters. A higher shortwave radiation increases near-surface water temperatures and can promote stratification, while a higher wind speed has largely the opposite effect. The effect of wind on mixing dynamics is notably different, so that given enough observations, the influence of the two variables can be separated. We could see that with lakes that had

longer time series of observed water temperature, the interaction measure was generally lower for GLM, GOTM, and Simstrat (Figure S7 in the Supplementary material). Though for the simpler hydrodynamic equations in FLake, separating the impact of wind speed and shortwave radiation seems to be more difficult. Similarly, the lake archetype seemed to influence the degree of interaction as well, perhaps extending to other parameters than meteorological scaling factors, which is in line with the findings of Andersen et al. (2021).

The sensitivity analysis and cluster analysis could provide hints towards improving global simulations without the need for model-specific calibration. The sensitivity analysis suggests that with parameter values ranges used here, the meteorological forcing data is the most influential in reproducing observed lake water temperatures. Comparing the distributions of best performing parameter values between the lake cluster gives indications on how to scale meteorological forcing (and potentially other, less sensitive parameters) for certain lakes, which could result in an overall improvement in simulating global lake

water temperatures. For instance, this lake data set and models showed clear improvement in model performance when scaling shortwave radiation and wind speed (Figure 7). Sheltering and the cubic scaling of wind speed with mixing may account for some of the need to scale wind speed, whereas the scaling of shortwave radiation is less easily explained, though heat



transport into the water column, shading, or another lack or excess of heat input may play a role. Regardless, an open question remains whether using results of the cluster analysis to parametrize uncalibrated simulations should be done. A clear weakness
of this study is the low sample size in some lake cluster (i.e. n = 3 for "deep" lakes). Also, the model configuration can be problematic, as for instance the influential `k_min` parameter in GOTM had strong effects on mixing and would therefore interact with meteorological scaling factors. Additionally, gridded data are supposed to give the best possible estimate of meteorological variables in a certain grid cell. Unless it can be shown that such data are skewed in a predictable way for lakes in particular, an adjustment of meteorological variables would mostly be needed to compensate for current sub-optimal process
descriptions in lake models themselves. So taking the above weaknesses into consideration, these findings raises the question: Is gridded forcing data adequate to replicate lake-specific meteorological conditions and thus can be used to reproduce lake thermal structure? And if so, should improvements in current model performance be found solely in improving hydrodynamic process descriptions?

## 5   Conclusions

We calibrated four different hydrodynamic lake models to 73 lakes using bias-corrected reanalysis data as forcing, and estimated the sensitivity of the calibrated parameters. From the six parameters calibrated for each model, only 2-3 were sensitive. This suggests that it can be sufficient to calibrate the models using only a subset of parameters. We achieved good model performances compared to previous studies and underscored that while some of the models were performing better overall, each model outperformed the others in at least some lakes. We analyzed four different model performance metrics and for
over 90% of all lakes, at least two models performed best for different performance metrics. To understand the effect of lake characteristics on the model performance, we grouped the 73 lakes into five cluster representing lake cluster. We highlight that both model structure and lake archetype were influencing model performance. In general, the three 1D models (GLM, GOTM, Simstrat) performed better than the 0.5D model (FLake). More specifically, Simstrat performed better at simulating water temperature at the depth of the thermocline than the other models. We attribute this to the seiche module included in
Simstrat. From these findings we conclude:

1. There is still room to improve model structure and process description of the 0.5D/1D hydrodynamic models. Especially (better) representation of deep-mixing processes, e.g., internal seiches, could potential benefit simulations results.

2. Using an ensemble of multiple hydrodynamic models is beneficial, especially as for these 1D (0.5D) models the computational cost of using multiple models simultaneously is low. But, there is still room to further take advantage of the
ensemble approach, e.g. by exploring weighted ensemble averaging techniques.

3. Even though we saw patterns in the best performing parameter sets regarding the lake cluster, it is unclear if using this approach might improve uncalibrated simulations (i.e. simulations where no observations are available). This is mainly caused by the fact that we used gridded forcing data, and meteorological scaling factors (wind speed, shortwave



radiation) was most influential on lake thermal structure which probably represent the importance of local orography and potential sheltering.

These conclusions serve as a baseline for understanding model sensitivity, and can support further improvements and developments of water temperature simulations and, thus, a better assessment of global change in lakes and reservoirs. Additionally, these conclusions can be the basis of a broader discussion about model uncertainty — especially when using gridded forcing data — and its relation to model design and parameterization.



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

*Acknowledgements.* This work was conceived at the Global Lake Ecological Observatory Network (GLEON), and benefited from continued participation and travel support from GLEON. We would like to thank Muhammed Shikhani and Lipa Gutani T. Nkwalale for valuable feedback on an earlier version of the manuscript and Tadhg Moore as well as Thomas Petzoldt for fruitful discussions during the study. JF received funding from the BMBF project FKZ 01LR 2005A—Fördermaßnahme 'Regionale Informationen zum Klimahandeln' (RegIK-605 lim). JPM was funded by the European Union's Horizon 2020 research and innovation programme, under grant agreement No. 101017861 (SMARTLAGOON). TKA was funded by the Grundfos Foundation (Lake Stewardship III).