# Peer review of "Learning from a large-scale calibration effort of multiple lake temperature models"

_EGUsphere, 2024_

## Author Comment (AC1)

We would like to thank the community member for their comments. We included the original comment in black font and **our response in bold violet font**.

Any planned changes or additions to the text are in violet font with boxes around them.

**Response CC1 – John Ding**

Figure 2. Distribution of the six evaluated performance metrics

In Figure 2, the histograms of the NSE (Nash-Sutcliffe efficiency) and R (Pearson correlation coefficient) highlight calibration results of four lake models on a daily step. The former is a variance-based, and the latter, a correlation-based metric. The simulated data counts cluster around a median value of 0.96 (an exact value not shown in the text) and of 0.98 (shown in Line 220), respectively, both at the upper end of their performance scale.

In a different context of rainfall-runoff modelling, for an observed hydrograph, Duc and Sawada (2023, Equation 25, Figure 2) show that the upper end/bound of the NSE is related to the correlation coefficient, R, as follows: NSEu=2-1/(RxR).

The median NSE and R values for simulated lake water temperatures appear to follow the equality.

Reference

Duc, L. and Sawada, Y.: A signal-processing-based interpretation of the Nash–Sutcliffe efficiency, Hydrol. Earth Syst. Sci., 27, 1827–1839, https://doi.org/10.5194/hess-27-1827-2023, 2023.

**Thank you for this interesting comment. It is reassuring to see that the model performance for the single best model is following the relationship. If you or anyone else is interested in further exploring the relationship between different performance metrics we would like to point out that the data containing all evaluated model performance metrics for the 2000 parameter sets for each lake and model are available in the Zenodo repository linked in the "Code and data availability" section of the manuscript (https://zenodo.org/doi/10.5281/zenodo.13165427). Below, we attached R code snippets to visualize some of these data. Please note that during the calibration we calculated NSE and Equation 25 in Duc and Sawada (2023) is using NSEu, the upper limit for NSE when there is no bias in the simulation, which might explain the difference. As these details are not strongly related to the main topic of the manuscript, we will not adapt any changes regarding this in the manuscript.**

```
############################################################################
* * *
**A short script to investigate the relationships between different model**
**performance metrics used in the calibration of 4 1D hydrodynamic lake**
**models. The data can be found in the results_lhc.zip file here:**
**https://github.com/aemon-j/isimip-sensitivity-analysis/blob/main/data/**
**Info on the calibration setup and the used models can be found in the**
**manuscript: https://doi.org/10.5194/egusphere-2024-2447**
**Author: J. Feldbauer, date: 2024-10-17**
* * *
############################################################################

**load necessary libraries**
library(tidyverse)

**read in data from the calibration**
res <- read.csv("results_lhc.csv", sep = ",", header = TRUE)

**filter the model runs to runs with OK performance metrics**
res <- res |> filter(abs(bias) <= 1, nse >= -2, r >= 0.6)

**data.frame with relationship (NSEu=2-1/(RxR)) from  Duc and Sawada (2023)**
dat_das <- data.frame(R = seq(-1, 1, by = 0.05),
                      NSEu = 2 - 1/seq(-1, 1, by = 0.05)^2)

**plot NSE against R, color code the points according to the bias and facet the**
**plot to the four used models. As for some parametrizations the NSE is very low**
**the y axis is limited from -2 to 1. Add line with NSEu=2-1/(RxR) relationship**
p1 <- res |> ggplot() +
  geom_point(aes(x = r, y = nse, col = bias)) + facet_wrap(~model) +
```

```
   geom_line(data = dat_das, aes(x = R, y = NSEu), lwd = 1.3) + xlim(0.6, 1) +

  ylim(-2, 1) + scale_color_gradient2(low = "red4", mid = "cyan", high = "red4")

**save plot as png figure**

ggsave("r_nse_rel1.png" ,p1, width = 10, height = 6)
```

---

## Author Comment (AC2)

We would like to thank the reviewer for their comments. We included the original comment in black font and **our response in bold violet font**.

Any planned changes or additions to the text are in violet font with boxes around them.

**Response RC1 – Zeli Tan**

Feldbauer et al. present a timely and very interesting study related to the uncertainty of physical lake models. Given the critical role of the studied models in climate impact analysis, such as ISIMIP, this research provides important implications for a better assessment of global change in lakes and reservoirs. The manuscript is with high quality of presentation and rigorous scientific inquiry. Only some clarifications and extended discussions are needed before it can be accepted for publication.

**We thank the reviewer for the positive words and we reply to the specific comments further down.**

One limitation of the current approach, which simultaneously assesses three types of uncertainties in lake models (i.e., input uncertainty, parameter uncertainty, and structural uncertainty), should be further discussed. Due to the interactions among these uncertainties, it is challenging for this multi-objective approach to fully resolve specific uncertainties. As admitted in the manuscript, the uncertainty of the three input-related scaling factors may hide the uncertainty of model-specific parameters and process descriptions. Consequently, the method obscures the investigation of an optimal model for specific lake types and an optimal algorithm for specific lake processes. Notably, for global lake modeling, we hope that as climate data become more accurate with time, the uncertainty of global lake simulations will be reduced. We also hope that lake models can achieve consistent global simulations without lake-specific calibrations. Despite the great values of the current paper, it falls short in addressing these issues. Conversely, I would encourage the authors to conduct a future study which uses observed atmospheric forcings to exclude the uncertainty in input data and ensure the good model performance achieved for right reasons. There are some existing studies following the recommended approach, such as Guseva et al. (2020) and Guo et al. (2021). But their values are limited due to either focusing on only one lake or testing only one lake model.

Guo, M., Zhuang, Q., Yao, H., Golub, M., Leung, L. R., & Tan, Z. (2021). Intercomparison of thermal regime algorithms in 1-D lake models. Water Resources Research, 57, e2020WR028776. https://doi. org/10.1029/2020WR028776

Guseva, S., Bleninger, T., Jöhnk, K., Polli, B. A., Tan, Z., Thiery, W., ... & Stepanenko, V. (2020). Multimodel simulation of vertical gas transfer in a temperate lake. Hydrology and Earth System Sciences, 24(2), 697-715.

**We agree with the overall statements that the reviewer makes, and that looking at multiple aspects of uncertainty makes it harder to focus on each individual contribution. Our reason for including input scaling factors was the considerable uncertainty that is present in these variables (i.e., meteorology and water**

transparency) for global simulations. The reviewer suggests an alternative approach for a future study. We wholeheartedly support such an idea, but this is currently infeasible with the ISIMIP data. As local meteorological forcing was not supplied and is likely not available for all sites, the bias-adjusted reanalysis data used here was the best available option for modeling. The proposed approach by the reviewer would therefore require a considerable data collection effort. Though we agree with the reviewer that this could be worth it, as observed meteorological data strongly reduces input uncertainty. We would raise the point, however, that even locally-observed meteorological forcing does not fully exclude uncertainty about the input data, as 1D models integrate signals from the entire lake and even meteorological observations in the center of the lake may not be representative of what the whole lake experiences. This is especially true for wind speed, which can have large temporal and spatial variations.

In L. 356-357, we already outlined the added benefit of using local observations. We want to modify this statement to incorporate some of the reviewer's suggestions:

> In a setting with locally observed meteorological forcing data, the model-specific parameters might become more influential, if meteorological forcing variables can be more constrained. Previous studies used this approach in one or a few lakes (e.g. Guseva et al. (2020); Guo et al. (2021)), but it would be beneficial to compile such data for a larger number of lakes, similar to the present study. Reducing the strong influence of meteorological scaling factors could facilitate identification of optimal models for different clusters. If observations are not available, improvements in downscaling methods from global products to weather conditions at the lake surface might also partially achieve this.

We will add a section to the discussion highlighting the reviewer's main remark:

> The overall uncertainty of mechanistic simulations is usually related to uncertainty in the initial conditions, uncertainty in the driving data (both forcing data such as meteorology and data used for calibration such as water temperature), uncertainty in the model parameter values, and structural uncertainty in the process description also called epistemic uncertainty (Thomas et al. 2020, Scavia et al. 2021, Dietze 2017). In this study we wanted to explore the relationships between lake model performance, parametrization, and lake characteristics, so we are mainly concerned with the uncertainties related to parameter values and model structure. The uncertainty in the meteorological forcing is thereby partly acknowledged by the inclusion of the scaling factors. Because the scaling factors proved to be amongst the most sensitive parameters, they could potentially prevent identification of an optimal model or patterns relating the parametrization of the models to the lake characteristics, if such an optimal fit exists. A way forward could be to reduce the uncertainty in the meteorological forcing data, and hence hopefully the sensitivity of the scaling factors, by using local meteorological observations instead of reanalysis data.

**Literature:**

Guseva, S., Bleninger, T., Jöhnk, K., Polli, B. A., Tan, Z., Thiery, W., Zhuang, Q., Rusak, J. A., Yao, H., Lorke, A., & Stepanenko, V. (2020). Multimodel simulation of vertical gas transfer in a temperate lake. Hydrology and Earth System Sciences, 24(2), 697–715. https://doi.org/10.5194/hess-24-697-2020

**Scavia, D., Wang, Y.-C., Obenour, D. R., Apostel, A., Basile, S. J., Kalcic, M. M., Kirchhoff, C. J., Miralha, L., Muenich, R. L., & Steiner, A. L. (2021). Quantifying uncertainty cascading from climate, watershed, and lake models in harmful algal bloom predictions. Science of The Total Environment, 759. https://doi.org/10.1016/j.scitotenv.2020.143487**

**Thomas, R. Q., Figueiredo, R. J., Daneshmand, V., Bookout, B. J., Puckett, L. K., & Carey, C. C. (2020). A Near-Term Iterative Forecasting System Successfully Predicts Reservoir Hydrodynamics and Partitions Uncertainty in Real Time. Water Resources Research, 56(11). https://doi.org/10.1029/2019WR026138**

I suggest the authors to avoid the use of "hydrodynamic lake models" to describe the studied models. To me, hydrodynamic models refer to numerical models that solve the transport of both mass and momentum. The authors can use "physical lake models", "thermodynamic lake models", or just "lake models".

**It is up for debate whether the processes described by these one-dimensional models are part of "hydrodynamics" or not; for GLM, GOTM, and Simstrat, we would argue this is the case, but we acknowledge that the FLake model simplifies many hydrodynamic processes. To avoid confusion about this terminology, we will refer to all models as "(process-based) lake temperature models".**

**We will change the usage of the term in the manuscript accordingly throughout the manuscript.**

In the methodology, one area that needs clarification is what procedure the authors have adopted to ensure appropriate initial conditions for simulations. It can be particularly important for the modeling of deep lakes.

**Yes, we had previously not provided this information. To give more information, we will append Section 2.1 with the following paragraph:**

> Initial conditions were estimated from observed water temperatures. Therefore, all available data in a period of days (depending on data availability) before and after the start date of the simulation were taken and averaged to set the initial temperature profile. All simulations used a spin-up period of 1 year.

**The scripts to set up the calibration are linked to in the Code and data availability section of the manuscript (https://zenodo.org/doi/10.5281/zenodo.13165427).**

**Minor comments:**

L55: also Zhuang et al. (2023). Zhuang, Q., Guo, M., Melack, J. M., Lan, X., Tan, Z., Oh, Y., & Leung, L. R. (2023). Current and future global lake methane emissions: A process-based modeling analysis. Journal of Geophysical Research: Biogeosciences, 128, e2022JG007137. https://doi. org/10.1029/2022JG007137

**We thank the reviewer for this relevant reference and we will add it to the examples.**

L181: gets absorbed

**We will revise this.**

L202: How does the metric δ differ from the Sobol's total-order index?

**Delta moment-independent measure should not be classified as a total-order index. We plan to correct this throughout the manuscript. Delta (δ) represents the importance of the entire distribution of the specific model parameter with respect to the entire distribution of simulated water temperatures (Plischke et al., 2013; Borgonovo, 2007; both cited in the main text). In contrast, Sobol' S1 estimates a parameter's influence on the variance of the simulated water temperatures. In this analysis, delta measures provide a second estimate to complement the variance-based Sobol first-order index to strengthen the analysis. As this study is interested in identifying the most important parameters (i.e. factor prioritization setting), we follow the recommendations of Borgonovo et al. (2017; cited in main text) and use both variance-based and moment-independent measures. We will change the description of both measures (from line 202):**

The analysis calculates two sensitivity measures, the moment-independent δ and variance-based Sobol S1. The delta moment-independent measure δ considers the influence of the entire distribution of a model parameter with respect to the entire distribution of simulated model output, whereas the variance-based first-order sobol index S1 calculates a parameter influence on the variance of the simulated model output (Plischke et al., 2013; Borgonovo, 2007). As this study was interested in identifying the most important parameters (i.e. factor prioritization setting), we followed the recommendations of Borgonovo et al. (2017) and used both variance-based and moment-independent measures to increase the robustness when estimating the inference of which parameters are most important when simulating water temperatures.

**And from line 251:**

From the calibration runs using the latin hypercube approach, we calculated moment-independent measure δ and variance-based first-order measure S1 for each combination of models, performance metrics, and lakes (Figure 5).

**To take into account parameter interactions, we also calculate S_interactions and discuss its importance for simulating water temperatures.**

L212: What is "SA"?

**We meant "sensitivity analysis", but this was an oversight and we will now write "sensitivity metrics" instead.**

L226-227: I suggest moving Figure 3 upward to Section 2.2

**We will follow this suggestion, as this would give readers an overview of the lakes we simulated early on in the text.**

L232: Figure S5 is introduced prior to that of Figure S4.

**We thank the reviewer for noticing this and we will switch the order of the two figures.**

Figure 5: It is surprising to see that S1 is larger than δ in many cases. To my experience, the first-order sensitivity should be smaller than the total-order sensitivity.

**We appreciate the reviewer's keen eye and the opportunity to strengthen our description of the sensitivity analysis metrics used. Please see our comments and revisions above (to the comment on delta and Sobol measures). In addition, it is not uncommon to see Sobol S1 values larger than delta values in the sensitivity analysis literature.**

L306: remove "a"

**We will revise this.**

L407: potentially

**We will revise this.**

---

## Author Comment (AC3)

We would like to thank the reviewer for their comments. We included the original comment in black font and **our response in bold violet font**.

Any planned changes or additions to the text are in violet font with boxes around them.

**Response RC2 – Fabian Bärenbold**

General comments

One-dimensional physical lake models are a widely used tool in simulations of lakes and reservoirs for diverse goals like now-casting, forecasting, or to compute mixing for use biogeochemistry. Although many different models exist and several of them are very widely used, to my knowledge no consistent comparison between them exists until now on a wide range of lakes. In general, research about the link between lake types, model parameters and uncertainty is not widespread and I think this manuscript is a good contribution to fill this gap. The manuscript is well organized and written with only few spelling mistakes. However, I think that there is a problem with one of the calibrated parameters of GOTM and not enough details on the observational data.

**We thank the reviewer for the positive feedback and helpful comments. We address the concerns below.**

Specific comments:

The observational data used to calibrate the 73 lake models is of great importance, but very few details are given in this publication. I would find it useful to a small paragraph about time and depth resolution of the observational data and whether any weighting was done to compute performance metrics (lines 196 – 198). Is there a minimum requirement of observations to be included to the 73 lakes? In addition, I would welcome a Figure giving some information about depth and time resolution of measurements in the supporting information.

**We acknowledge that we provided too little information about the observational data. We will modify lines 64-65 to:**

The resolution (vertical and temporal) of the observed data and the detail of the hypsograph varied per lake. For all but two lakes, data covered a period of at least 1 year, and for 75% of the cases it covered at least 5 years. Profiles (three unique depths or more) were provided for all but four lakes, and all lakes had more than 100 unique observations (Figure <X> in the Supplementary material). A link to the observed data and hypsographs is provided in the Code and data availability statement.

**The available data varied quite a lot, so there would often be one or two exceptions to general statements about the data. We will provide a link to the observed data, as made available to ISIMIP modelers, in the Code and Data availability statement: https://github.com/icra/ISIMIP_Local_Lakes/tree/main/LocalLakes.**

**We agree with your suggestion and we will add a figure to the Supplement:**

[Figure]

**The very high number of unique depths in observation records are caused by the fact that some of the observational data were from CTD profiles, which we aggregated to a vertical resolution of 0.1 m before plotting. This information will be included in the figure text.**

**Weighting was not performed. We will modify L. 196-198 to state that the metrics were calculated on all water temperature observations:**

> We sampled and ran the four models for 2000 parameter sets, and for each of the parameter sets we calculated four performance metrics over all water temperature observations: Root mean squared error (RMSE), Nash-Sutcliffe model efficiency (NSE), Pearson correlation coefficient (R), and mean error (bias)

Is there a reason the GSWP3-W5E5 reanalysis dataset was chosen? Could you explain this in 1 – 2 sentences? Also a bit related, is there any chance of going to hourly instead of daily resolution? As far as I understand from the discussion this might solve some of the problems (wind speed effect on mixing is cubic). If yes, it would be interesting to mention this in the discussion/conclusion.

**The GSWP3-W5E5 dataset was chosen because one of the aims of the calibration was to run the ISIMIP3 local lakes climate simulations, and W5E5 was used for the bias-correction in ISIMIP3. These climate simulations are not part of this paper, but can be found on the ISIMIP data portal (linked to in the Code and data availability statement). We now briefly mention the aim of the calibration in the text, but we reject**

**adding a long explanation in order to not confuse the reader by referring to simulations that are not part of the paper.**

**After L. 45, we plan to add:**

These calibrations were in preparation for ISIMIP climate impact simulations for the local lakes sector (see Code and data availability).

**The effects of going from daily to hourly resolution are interesting. To our knowledge, there are no hourly meteorological forcing data available in ISIMIP3 at the time of writing. However, temporal downscaling could be considered to take into account the importance of diel variations, as the reviewer is suggesting. This was done in a previous study by Ayala et al. (2020, cited in main text) using local observations and artificial neural networks. Their results suggested indeed an improvement when using hourly data compared to daily, although improvements when using synthetically-generated hourly data were minor. Relating to the reviewer's comment about wind speed: their calibrated wind scaling factor was similar (1.5) when using daily or synthetic hourly, while it was lower when using observed hourly forcing (1.4). We expect that when focusing on general physical trends and long-term changes in water temperature, daily forcing is sufficient. However, when simulating short-term events or extending to biogeochemical simulations, sub-hourly forcing will become more valuable or even necessary.**

**To mention the effect of a higher temporal resolution in the text, we plan to append the text after L. 357:**

Similarly, use of hourly meteorological forcing could result in more realistic patterns in wind-driven or convective mixing (Ayala et al., 2020).

I would find it interesting to have the silhouette plot of the cluster analysis in the supporting information.

**We plan to append the silhouette plot to the supporting information. As stated in the manuscript we manually chose to use 5 lake clusters even though the silhouette plot suggested the optimum number was 2, because we are convinced that it is more informative. To arrive at the 5 clusters we tried out different numbers of clusters (4, 5, 6) and decided that 5 was the most informative (i.e., containing most scientifically interesting limnological characteristics for discussion) while not creating too many clusters with very few members.**

[Figure]

The authors discuss the fact that k_min of GOTM seems to be the most sensitive of the lake-specific calibration parameters. The range of k_min used in this study is $1.5e^{-7}$ to $1e^{-5}$, which is rather high values compared to a default value of $1e^{-8}$ in the GOTM manual. In addition, typical values of $\sim 1e^{-6}$ are reported for TKE in the hypolimnion of lakes (e.g. Wüest and Lorke, 2003). I see a major problem if k_min is set too high: it could, together with high values for the scaling of shortwave radiation, offset a low value for the scaling of wind speed. There is evidence of this in Figure 7, where the calibrated parameters for GOTM are consistently low (wind speed) or high (shortwave) compared to the other models. The calibrated value of k_min always seems to be well above $1e^{-6}$ (Figure S8), so on the order of typically observed values for TKE (Wüest and Lorke, 2003). The authors discuss the potential interactions between k_min and wind scaling on one hand (lines 341 – 345), and shortwave scaling and wind scaling on the other hand (lines 365 – 369) but not the potential interaction between all 3. In regard of this, I would like to ask the authors to motivate their choice of the range of k_min and to check its influence on the wind and shortwave scaling parameters. I see two ways to do this:

- Compute second/third order Sobol indices for the concerned parameters
- Redo some of the simulations with k_min = 1e-8

I understand that the interaction calculation shows that this suggested parameter interaction is not driving variance but the correlation could still be strong among these 3 parameters. There is also no obvious reason why wind and shortwave scaling should be so different for similar models like Simstrat and GOTM. I could be wrong but to me it seems like k_min in GOTM is playing the role of the seiche in Simstrat.

**We chose the range for the calibration of k_min based on past experience and discussion with colleagues experienced in  GOTM (e.g. Andersen et al. (2020) & Ayala et al. (2020); both cited in the main text). We believe that the default value for k_min from the GOTM manual is for the ocean as we are not aware of a reference manual for**

the lake branch of GOTM. As mentioned in the reviewers comment, typical values for TKE in lakes are larger. We got feedback from the GOTM developers that: "k_min is 'un-resolved TKE generation' and should in principle cover seiching. You write that k_min value varies a lot between lakes, which could be a result of the importance of seiching for a given lake." (Personal communication with Karsten Bolding, 2024).

We further investigated the calibration results and found that the k_min values for the best performing parameter set are larger for deep, medium temperate, and warm lakes compared to small temperate lakes (see attached figure below, subpanel A). The exceptions are the large shallow lakes that cover a wide range of k_min values. This could be explained by looking at the sensitivity of k_min for the different clusters, whereas for the large shallow lakes k_min has very low sensitivity values (Figure below, subpanel B).

[Figure]

We do agree with your comment regarding potential interactions or correlations between swr scaling, wind speed scaling, and k_min. So we followed your suggestion and ran an alternative round of calibration for GOTM with k_min fixed at 1e-8. In addition, we ran an additional round of calibration for Simstrat with a_seiche set to 0, which disables the seiching module for Simstrat. This was suggested by a colleague to test if the improved performance of Simstrat was caused by the inclusion of seiching as we speculate in the discussion and allows us to better compare the impact of reducing k_min (which as mentioned above should also cover seiching). Under these additional calibration rounds (GOTM with k_min = 1e-8 and Simstrat with a_seiche = 0) both of the models showed worse performance in most of the lakes, especially for the medium temperate and warm lakes. For large shallow lakes, very small differences were seen and for some of the small temperate lakes we even saw better performance for both Simstrat and GOTM (See attached figure below; the difference in the plot is calculated as old calibration - new calibration).

[Figure]

**However, as the reviewer suspected, with k_min kept at 1e-8 the calibrated wind speed scaling from GOTM (GOTM_r in the attached figure below ) increased and became more similar to the other models.**

[Figure]

**For the calibrated swr scaling we do not see a reduction in the values for the best performing parameter set for the new round of calibration (GOTM_r in the attached figure below). The calibrated values even slightly increase when keeping k_min at a constant value of 1e-8.**

[Figure]

Reducing k_min reduced the overall model performance of GOTM for the lakes where deep mixing (incl. internal oscillations) is of importance (as seen by the similar reduction in model performance of Simstrat when a_seiche = 0). In GOTM, increasing wind speed scaling factor can (to some degree) compensate for this, but it is not able to perform nearly as well as with larger k_min values. As GOTM cannot reach similar performance by increasing wind speed scaling and reducing swr scaling, we suggest that there is potentially no strong interaction or correlation between the three parameters (as previously seen by the sensitivity and interaction measure). Only for some of the small temperate lakes, a lower k_min value (and a_seiche = 0 for Simstrat) is resulting in better model performance.

We will add a paragraph reflecting this discussion in the supplementary information. Also, we plan to update all scripts and add the new calibration data to our github and the Zenodo repository

In L324 we plan to add

> We reinforced this hypothesis by performing additional simulations with a_seiche set to 0, which lead to poorer model performance of Simstrat (see supplementary material for details).

In L340 we plan to add

> for all lake clusters besides the large shallow lakes (Figure <Y> in the Supplementary material).

In L386 we plan to add

> (more details on this can be found in the supplementary material).

The authors seem to imply that larger lakes should generally have a larger value for wind scaling and vice versa for small lakes (lines 349 – 351). However, if true this effect should be visible in the calibrated wind scaling parameters, no?

**The mentioned implication was our initial hypothesis, but our study did not find a relation between best parameter values for wind speed scaling factor and lake size to confirm this. A possible reason for this is the resolution of the gridded data. In order to clarify this, we plan to add a sentence:**

> We could not highlight any relations between best parameter values for wind speed scaling factors and lake size, which could imply the gridded weather data mask any effects of lake size.

Technical comments:

Title: maybe "hydrodynamic" or "physical" lake models instead of just lake models

**We agree this title was not restrictive enough. We will modify the title (also in connection to RC1's comment) to:**

> Learning from a large-scale calibration effort of multiple lake temperature models

Line 6: The following sentence is a bit too unspecific to me: "The models performance and parameter sensitivity showed a relation to the lake characteristics and model structure."

**We agree and will modify the sentence accordingly:**

> Parameter values, model performance, and parameter sensitivity differed between lake models and between clusters that were defined based on lake characteristics.

Line 39 - 40: Maybe mention that although important for shallow lakes, the biogeochemical components are not discussed in this manuscript.

**We agree that it was unclear if we were talking about our study or Andersen et al. (2021). We will modify L. 37-40 as follows:**

> Andersen et al. (2021) performed an extensive, global sensitivity analysis on the 1D coupled physical-biogeochemical model GOTM-WET in three Danish lakes and found that parameter sensitivity may be strongly linked to lake morphology, including a potential

feedback of biogeochemical components on temperature (such as light absorption by organic matter) in shallow lakes.

Line 64: Make clear that Table S1 of the current manuscript is meant and not Table S1 of Golub et al. (2022)

**We will change this to make it more clear to the reader by citing the Golub et al. paper in the previous line.**

Line 82: delete "and" before "the 1D turbulence-based models GOTM"

**We will revise this.**

Line 99: "comma" after "Equation 1"

**We will revise this.**

Equation 3: Shouldn't it be h < z < D_lake?

**Yes, thank you for noticing this. We will revise it.**

Equation 5: Please check whether term 1 is really negative.

**Yes, the first term on the right-hand side needs to be negative to account for the vertical gradient of short–wave radiation to act as a heat source for the water column with the underlying assumption that the depth dimension is positive (reference at the surface).**

Line 220: This sentence is not clear to me. Did the authors intend to say "between best and worst performing model"?

**Yes that is what we wanted to say. This will be revised.**

Line 263: "were" instead of "was"

**We will correct this.**

Line 269: "Lake clusters" instead of "lake cluster"

**We will correct this.**

Line 357: "better" instead of "more"

**We will correct this.**

---

## Author Response (AR1)

We would like to thank the editor for their comments. We included the original comment in black font and **our response in bold violet font**.

> Any planned changes or additions to the text are in violet font with boxes around them.

**Response to editors comments**

Dear authors,

Thank you for the detailed and serious responses to the questions raised by the reviewers.

**Thank you for a timely review process, our paper has definitely been improved by the reviewers and your comments.**

It remains a bit unclear to me what happen when the "unresolved TKE" expressed in k_min is larger than the "resolved TKE". Do we need to have a turbulence model in such case? I agree that it is important to add more details and perhaps open questions into the SI

**In GOTM, if the TKE calculated from wind shear is lower than k_min, TKE is set to k_min, so that at any time TKE >= k_min. We added a sentence to the methods section explaining this:**

> In GOTM, whenever the simulated TKE is lower than the calibration parameter k_min, it is set to the value of k_min.

**As written in the response to the reviewers we added a section to the supplemental material discussing the results of the new calibration round (with reduced k_min and a_seich set to zero). We end the section with the open question why GOTM is behaving so different from the other models in regard to the calibrated scaling factors:**

> An open question remained as to why the values for the calibrated wind speed and shortwave radiation scaling for GOTM behave so differently compared to the other lake models (Figure S14 and S15). This is especially unexpected for Simstrat, which is the most similar to GOTM in terms of process description and even showed a similar reaction in the additional calibration round (reduced k_min and a_seiche = 0) where we saw decreased performance in some of the lake clusters (Figure S13).

**In addition, we made some small changes in the text to improve language, make the text more consistent in terms of the used terminology and units, and added two more references to calibration parameters in the methods section. All changes are highlighted in the marked-up manuscript version.**

I am looking forward to seeing the revised version

Best